# "Violent Times Call for Violent Prayers": "Divine Violence" during the COVID-19 Pandemic in the Mountain of Fire and Miracles Ministries, Nigeria

Benson Ohihon Igboin [1,2]

1    Department of Religion & African Culture, Adekunle Ajasin University, Akungba 342111, Nigeria;
     benson.igboin@aaua.edu.ng
2    Nigeria & Research Institute of Theology and Religion, University of South Africa, Pretoria 0002, South Africa

**Abstract:** Interest in studying prayer has significantly increased because of the belief that it helps humanity to cope, particularly in times of crisis. Prayer is not just a communication with God, it is also an instrument of bond and embodied ritual among prayer litigants or people who belong to the same religious community. This article argues that divine violence, a sovereign act of God, was crucially needed by the litigants in order to guarantee human flourishing in the face of existential threat. The article studied how violent prayer—a genre of prayer that is targeted at the spirits underlying physical manifestation of suffering, pain, or crisis—was utilised by the Mountain of Fire and Miracle Ministries (MFMM) in Nigeria to cope with the fear and uncertainties occasioned by the COVID-19 pandemic. This article is part of an ongoing ethnographic research on "The Politics and Poetics of Violent Prayer in the Nigerian Pentecostal Churches", which began in 2021. I utilised ethnographic engagement, particularly interviews and participant observation, to attempt to understand what these prayer litigants are doing when they pray violently.

**Keywords:** violent prayers; spiritual warfare; Nigerian Pentecostals; poetics; machine gun prayers; spiritual technology

## 1. Introduction

Several studies have shown a resurgence of prayer at the turn of the outbreak of coronavirus in December 2019 (Bentzen 2021; Nyamnjoh et al. 2022; White 2022; Tatala and Wojtasiński 2021; Szałachowski and Tuszyńska-Bogucka 2021; Mwale and Chita 2022). These studies focus on how people across the world deployed prayer as an instrument to contest, resist, succour, or overcome the threats of the COVID-19 pandemic. The emphasis on resurgence is imbued with the belief that prayers were waning in the period preceding COVID-19 or that prayer's intensity had reduced prior to the pandemic. The resurgence of prayer also challenges the facile notion of prayer that had become suffused as a result of nontheological or secular appraisal of prayer (Fitzgerald 2012). As my fieldwork has also confirmed, even though people gathered in great numbers to pray prior to the outbreak of the COVID-19 pandemic, the intensity with which they prayed during the pandemic could not be compared with their fervency before the pandemic. Although they did not gather in large numbers during the pandemic, Nigerian Pentecostals intensely prayed and trusted God to heal the world of the virus, thus introducing what they refer to as divine violence to counter the violent intrusion of the pandemic, hence violent prayers.

The broad aim of this article is to examine how the members of the Mountain of Fire and Miracles Ministries (MFMM) understood and deployed the concept of divine violence in their violent prayer mode during the pandemic. Generally construed, divine violence means that God is a sovereign being whose authority cannot be questioned. As such, when he (God) deplores violence to counter human or other spiritual violent intrusion into his creation, the victims of divine violence are neither regarded as saints nor is their death

perceived as a sacrifice (Zizek 2008). In the context of this article, the MFMM conceives of divine violence as supernatural intrusion and intervention in human existential crisis, which aims to destroy the spiritual forces perceived to be responsible for the outbreak and spread of the COVID-19 pandemic.

To be sure, violent prayer is a special genre of prayer that scholars have described as spiritual warfare prayer, machine gun prayer (Marshall 2016; McAlister 2016; Richman 2020), deliverance prayer, spiritual technology against spiritual forces (Adogame 2012), dangerous prayer, apocalyptic device, a prelude to the final Armageddon (Adelakun 2023), and so forth. This form of prayer was once an exclusive preserve of the Mountain of Fire and Miracle Ministries, Nigeria, from where other Pentecostal churches across Africa and beyond began to borrow (Adelakun 2022b). This genre of prayer is theologically located in the event of the Pentecost at Jerusalem, when "suddenly a sound like the blowing of a violent wind came from heaven and filled the whole house...where they [the apostles and others numbering 120] were sitting" (Acts 2: 1–2) (Oden 2011, p. 107). In addition, these Pentecostal Christians claim that Matthew 11:12 grounds them theologically to embark on violent prayer. For them, since the kingdom of God is taken by force, violent prayer becomes the instrument to achieve it. In reality, however, as a combative, aggressive, and confrontational prayer, violent prayer is targeted at spirits that are believed to fight against human existential progress or success. Dislodging or overcoming these spirits through violent prayers is believed by the litigants to be the way to personal freedom and a guarantee to enjoy material success (Butticci 2013). Violent prayer, as an apocalyptic device or machine gun, is also aimed at disestablishing powers and authorities both political and spiritual in order to establish an orderly, righteous reign. In this sense, violent prayer is subversive: this is its core (Adelakun 2023).

However, the suddenness of COVID-19 and its quick spread across the world violated the relative peace that was being "managed" by different countries worldwide at national and international levels. For the Christian world, the pandemic violated the tranquil or convivial nature of their communal worship and fellowship, a bond that makes the Christian (Pentecostal) community an embodied community. Pentecostal worship is stoutly expressive; it is embodied. Prophetically, at least, in Nigeria, the pandemic violated the discursive Pentecostal leaders' prophetic or spiritual antennae because they rarely saw it coming. This is profound because every New Year's Eve was usually characterised by volumes of prophecies of what supposedly lay ahead. Even though the prophecies have raised some credible concerns, and some scholars are already interrogating them (Ugwu 2009), the COVID-19 pandemic violated the (dis)order of prophetic rendition in Nigeria.

The COVID-19 pandemic also violated the bond between wealth and health teaching: isolating wealth and emphasising health although momentarily, the Pentecostal leaders quickly realised the existential urgency that violent times have placed on surviving the pandemic first of all (Igboin 2022a, 2022b). The theology of hope in crisis, faith in uncertain times, courage in depressing moments, love in obscure circumstances, and so forth became resonant in the e-congregation and small fellowship centres that characterised the times of the lockdown (White 2022). This is not to suggest that there were no attempts at demonstrating the usual boisterous flouting of wealth by a small number of Pentecostal leaders, but the majority of them certainly and soberly comported themselves with reserved demeanour (Igboin 2022a). The COVID-19 pandemic also practically violated the secular space and trudged both the spiritual and secular spaces with anxiety (Ukah 2020).

The COVID-19 pandemic violated the space in terms of its suddenness and interruption of normalcy enjoyed before its outbreak, introducing the concept of the "new normal". The new normal forced Nigerians and the rest of the world to appreciate the bond they had taken for granted for so long that they were compelled to resurrect, rest they so desired, which the pursuit of material success had taken away from them (Zizek 2020); deep human and spiritual reflection on who they are and the purpose of their existence, which modernity, and biomedicine with their disenchantment had almost completely robbed them of (Reimer-Kirkham et al. 2020); the near collapse of class, identity and status (equality of

humanity at least during the lockdown), which they had often advocated but rarely lived; the need for survival and longevity, which advancement in medical science and practice had assured them was guaranteed (Igboin 2022c); unmitigated ecological violence, which they had caused with unease pleasure from industrialisation and technologisation; the soul of humanity, which they had taken to a robotic realm suddenly reincarnated, at least, for a moment (Knibbe 2020); desertion of the grandiloquent and aesthetic cathedrals where they once believed that God usually dwelt in and bonded with them in human form (Adelakun 2022a); and so forth.

The uncertainty and the imminent fear that the COVID-19 pandemic generated globally caused a different mode and mood of tension for and in Africa, and Nigeria in particular, because of the despicable nature of medical infrastructure. Inveterate corruption and leadership failure have negatively affected almost all aspects of the Nigerian government structure, and confidence in the system is almost completely lost. The predictions that the streets of Africa would be littered with human bodies added to the intensity of fear and need for prayer (Adelakun 2022a). Perhaps building on the experience of the Spanish flu of 1918–20 that tremendously affected Africa (Heaton and Falola 2014), Melinda Gates, for instance, predicted: "I am worried. . .. I see dead bodies in the streets of Africa" (Omanga and Ondigo 2020), while the International Director of SOS Children's Villages in Eastern and Southern Africa, Senait Bayessa, also predicted that "Africa will be the hardest hit by the impact of the outbreak". In addition, the United Nations Economic Commission for Africa (UNECA) predicted that there would be about 3.3 million deaths and 1.2 billion infections throughout Africa (Adebowale and Onyeji 2021). These predictions from developed countries where medical facilities effectively work were a source of great concern for Africa, generally, and Nigeria in particular. Despite the uncertainty and fear that pervaded society during the pandemic, many Nigerian Pentecostals found hope to live through the violent times by recourse to "violent prayers".

In this article, I examine violent prayers as a response to the "violence" occasioned by the COVID-19 pandemic in Nigeria with particular reference to the Mountain of Fire and Miracle Ministries (MFMM), Nigeria. In nuancing violent prayer, I will ground it on the concept of divine violence, which intrudes, intervenes, irrupts, and disrupts human processes and acts in ways that violent prayer litigants understand as miracles. I will elaborate on the existential, theological order and the poetics that ground these prayers, which became evident during the pandemic. I ask, what were Pentecostal litigants really doing when they prayed violent prayers during the pandemic? What are the theological potentials contained in violent prayers that can be fashioned to drive towards African Pentecostal theology of prayers in a crisis situation? To be sure, Christians who practise this genre of prayers use certain kinetic aesthetics through which they curate the thought that goes into this radical form of engagement, and they spontaneously perform them in their prayers and everyday actions. I employed ethnographic resources (interview and participant observation) to interrogate this form of prayers in times of crisis in order to understand what the prayer litigants did when they prayed violently during the pandemic, and how their action or performance (of violent prayers) generated the spiritual (faith) resources to cope and/or overcome the pandemic. Data collection for this study involved field observations and semi-structured interviews with nine members of the Mountain of Fire and Miracles Ministries in Lagos, Akure, and Akungba-Akoko, all in Nigeria, between August 2021 and September 2022. The data are part of the research project titled "The politics and poetics of violent prayer in Nigerian Pentecostal churches" sponsored by a grant awarded by the NAGEL Institute, USA. The research seeks to interrogate what the litigants do when they engage in violent prayers. The names of the respondents have been changed in order to preserve their anonymity.

## 2. Conceptualising Violent Prayers in Times of Crisis

There is a growing interest in the academic study of prayer in its theological, nontheological, and practical dimensions. In its theological and practical resonations, interest in

prayer has shown, among other things, how prayer is a resource not only to relate with otherworldly beings and seek their help or intervention but also to navigate and mediate existential challenges that confront prayer litigants daily. Human attitude towards God in times of crisis is different from times of ease or peace. Religion obviously plays a critical role in times of vulnerability (Stańdo et al. 2022). Bentzen (2019) points out that the intensity of prayer can be driven by personal, collective, or natural crises. Turning to prayer and other (spi)ritual activities in crisis stoke both emotional and spiritual feelings. Bentzen (2021) has further explored how COVID-19 aroused interest in prayer around the world. Bentzen unravels how people prayed for comfort, an end to COVID-19, and so forth. Exploring the spiritual turn in Southwestern Nigeria, Olonade et al. (2021) show the intensity of prayer to "ward off" the COVID-19 pandemic in the world. Yendell et al. (2021, p. 13) note that the rise and intensity of faith in prayer during the initial stage of the pandemic could have made "secular people" wonder if the act of praying was not "an expression of latent helplessness". But, as it turned out to be eventually, prayer played a prominent role in how people coped with the pandemic.

The question thus is, what is prayer? In their article, *"Yes, in Crisis We Pray"*: The Role of Prayer in Coping with Pandemic Fears", Szałachowski and Tuszyńska-Bogucka (2021, p. 1) ask "What is prayer in the Bible? It is how those who believe in God talk to him. That is how they reveal their eulogies and requests". The emphasis on "how" is at first intriguing because, primarily, prayer is conceived as a "what" (communion) between Christians and their God; it is, therefore, an act, a performance, rather than a method, means, or manner. But, on a deeper thought, it could also be correct to think of prayer as a method, means, or manner of reaching God. As a result, one can understand Szałachowski and Tuszyńska-Bogucka's definition of prayer as both a performance (act) and a method, means, or manner of demonstrating how prayer is offered or performed. In a rhetorical sense, Fitzgerald (2012, p. 2) states the following:

> Prayer as a discursive art in which capacities central to our human experience with language come together with respect to supersensory, superordinate, supernatural reality, typically imagined in the form of culturally significant otherworldly audiences—divine beings with whom human beings enjoy rich, complex relationships.

Fitzgerald does not only conceive prayer as a discursive art but he also nuances it as a performative, pragmatic art culturally conditioned but often tailored towards recognised and accepted divine beings, in whom humans repose and trust. Fitzgerald largely focuses on prayer as a rhetorical discourse, which does not align much with the theological understanding of violent prayer but on the analysis of its language and structure. He believes that the rhetoric of prayers will help us to appreciate the ontological and epistemological meanings imbued in prayer through the study of its language and structure. According to him, prayer can be conceived as "a techne for generating or maintaining presence" (Fitzgerald 2012, p. 58) of the divine being with whom the prayer litigants seek help. It must, however, be noted that the theological production of prayers is a veritable resource for the analytical turn that the rhetoric of prayer assumes and emphasises.

Theological production of prayer refers to the raw, embodied prayers, "vocal prayer, discursive prayer, the prayer of quiet, the prayer of full union, of ecstatic union, spiritual bouquets, spiritual marriage" (Casalddliga and Vigil 1994, p. 121), which litigants offer or engage in consciously or otherwise. This instantiates the fact that situation or context determines the kind, poetic, and performance of prayer litigants pray or are involved in. As Fitzgerald (2012, p. 15) himself acknowledges in *Spiritual Modalities: Prayer as Rhetoric and Performance*, "A critical moment calls for decisive judgment. . .. Experience of need opens one to prayer, and prayer becomes an experience of *blessing*" (emphasis on the original). As the criticality of one's situation and need for prayer unfurls, the push for a particular mode of prayer intensifies. Fitzgerald (2012, p. 29) adds that "there are far more situations for prayers than there are prayers for situations". Thus, the prayer we are examining here is

borne out of a (dis)pressed moment when the threat of existential survival was real and the need for immediate divine intervention was intense.

Of course, prayer's upward look for divine intervention suggests that the need being prayed for cannot be met immediately by human efforts or facilities. This is not to suggest that prayer is not a human effort. This accounts for why Reimer-Kirkham et al. (2020) conceptualise prayer as an experience that contradicts the rudiments of technologies, temporalities, and biomedicine. They argue that prayer is a transgressive art that challenges and transcends social rules, interrupts clinical machinery, ruptures established order, intersects and simultaneously penetrates secular and religious spaces, and interpenetrates the otherworldly realm. They posit that prayer is not authorised from the abstract realm, it is "shaped by the social and cultural contexts in which it occurs" (Reimer-Kirkham et al. 2020, p. 21). In other words, both individual and corporate prayers are initiated by the prevailing social and cultural exigencies, which require some form of divine intervention. Prayer thus becomes a channel or means to an end.

During my fieldwork in the Mountain of Fire and Miracles, I also observed that context and the prevailing situation play critical roles in authorising prayers and that prayers are a means to an end. In an interview, Dr. Daniel Olukoya (2013), the founder and presiding pastor of the Mountain of Fire and Miracles, told us that personal, social, cultural, and spiritual context or experience determines prayers and the particular kind of prayer. For him, the presence or abundance of social amenities does not count when one experiences a spiritual need for violent prayers. Acolatse (2018) reminds us that the Western theological context for prayer differs from that of Africa: rather than ascribe spiritual causation to mundane challenges as most Africans do, the West will direct their energy towards the socio-political causation of situation and actively work towards rectifying it. In other words, the same dysfunctional situation in the West and Africa is most likely to be interpreted and addressed differently: in the West, politically, while in Africa, spiritually.

Akadiri, a pastor at Mountain of Fire and Miracles, conceptualises prayer as a human–divine communion, which, more often than not, results in intervention in human situations and human favours. According to Akadiri, a particular situation births the kind of prayer that litigants pray. Since the situation determines the kind, mode, and mood of prayers, Akadiri argues that the prayers prayed during the outbreak of COVID-19 were first and foremost instigated by and targeted at the pandemic. In his words,

> The prayer we prayed during the pandemic was focused, and pointed at the termination of the pandemic because it was the pandemic that prompted it in the first place. It was also specific and timely. And we equally believed that since the moment was a violent one that threatened people's lives, the response should be violent because we wanted people to be delivered. . . . We prayed violently in order to put an end to COVID-19. . . .. There was divine violence that cancelled the violence of the pandemic.[1]

Akadiri also reveals that prayer generates unusual power, and it can be violent (hence violent prayers) when faced with a labyrinth of life or an existential threat, such as the one posed by the COVID-19 pandemic. More importantly, he raises issues of violent prayers and "divine violence" in times of crisis, which will be explored below. Adelakun (2022a) also observes that in Nigerian Pentecostalism, power plays a prominent role in mediating between existential and ultimate space, and this power can be obtained through fervent prayers. According to her, "Prayers are spiritual means of accessing power; the sheer physicality makes the body a channel to draw and exercise power" (Adelakun 2022a, p. 75), where "power is the ability to transcend the limit of one's human ability, and it is essential to relationship between humans and God" (Adelakun 2022a, p. 13). Performing power as an embodied act, according to Adelakun, is the hallmark of Nigerian Pentecostalism because, without power, Nigerian Pentecostalism would be dull, staid, and stale.

The performance of power is contextually authorised just as violent prayers are initiated by the prevailing situation in individual or community life. In my fieldwork, I observed that praying violently is grounded not only in an African spiritual worldview be-

lieved to be characterised by spiritual forces capable of adversely affecting human progress but also in the colonial experience of exacerbated violence that prompted violent struggles for independence. In addition, the military regimes most African countries experienced after independence added and extended the impetus of violence that colonialists deployed in Africa. The ingrained psychological fear of those periods has to be tackled and exorcised, according to the Pentecostals, through violent prayers. Thus, I conceive violent prayers not only in terms of exorcising spiritual forces but also as a means of self-decolonisation. This notion was explicitly explained by one of my respondents.

> You researchers always don't link violent prayer with colonial experience of violence. You need to study history. Africa was conquered by violence by the colonial masters. Apartheid in South Africa was a product of violence. It was also sustained by violence. I hope you know that as well? A violent situation naturally will make you violent. What you do is to do anything that you feel can make you escape violence. That is why there were several violent reactions to colonial violence. . .. While many Africans resorted to offer sacrifices to the ancestors against the colonial masters, Christians, particularly the African churches [African Independent/Initiated Churches] violently prayed to God against the colonial masters. That is the genesis of violent prayers in Nigerian churches today. You people [the researchers] need to know this (Interview with Ayeni 2021).[2]

Although it is difficult to ascertain how the African Independent Churches violently prayed against colonialism, it is trite to argue that these churches were the first to pragmatically take steps to free themselves from colonial churches (Igboin 2006). In reality, the Nigerian historical and political trajectories have, many Pentecostals suggest, not been free from crisis. The Nigerian Civil War (1967–1970) became a precursor for spiritual revival, just as the effects of the Structural Adjustment Programme of the 1980s to early 1990s helped Pentecostalism to maintain a steady presence in public life in Nigeria. The failure of the Nigerian state, inveterate corruption, deplorable economic hardship, and widespread poverty drove people into the Pentecostal churches where violent prayers against the spirit of poverty were literally dealt with (Obadare 2018; Igboin 2022a). In other words, the intensity and regularity of prayer is birthed and sustained by the continuous social, political, and economic crises that have stubbornly refused to be exorcised from the body politic.

However, in this present work, I concentrate on the spiritual intervention in human affairs that violent prayers are believed to initiate, particularly during the initial stage of the COVID-19 pandemic. But, a brief conceptualisation of violence is in order.

## 3. Conceptualising (Divine) Violence in Violent Prayers

Conceptualising violence in its entirety is a difficult task. This is because of the vast literature and temperaments that are involved in the art of violence itself. A generally accepted cliche is violence begets violence. While Raine (2013) grounds violence in biology, explicating how evolutionary accretions have engineered violence in humans, Grebrewold (2009) locates violence in social institutions, politics, culture, and so forth. Anyone who reads Frantz Fanon's ([1961] 2004) *The Wretched of the Earth* will appreciate the value, politics, and limitations of violence and its deployment in confronting an enemy in a difficult terrain of existential and political deliverance. According to Fanon ([1961] 2004, p. 51), violence is not an impotent weapon; it is, on the contrary, "a cleansing force", that helps the oppressed to pale "their inferiority complex", "restores their self-confidence", and "hoists the people up to the level of the leader". In addition, he states that "Violence can thus be understood to be the perfect mediation. The colonized man liberates himself in and through violence. The praxis enlightens the militant because it shows him the means and the end" (Fanon [1961] 2004, p. 44). For Fanon, by authorising and performing violence, the people unleash a totalising force that helps break the shackles of colonisation. Violence thus becomes a self- and collective decolonising tool that reaches the core of the oppressed and sutures the structures that hold the people bound.

The justification of violence as an art and instrument to confront and dislodge the colonists from further brutal subjugation of the colonised, and the virulent violence that it birthed in return has raised the question of whether violence can actually resolve human conflict. Hannah Arendt's (1970) reflections in her *On Violence* provoke analytical interrogation of violence. National and international violations and violence that characterised the twentieth century, she reasons, defy the political and economic categories that justified violence because at the end of it, what is gained cannot comparatively stand in balance with what is lost.

Beyond state or individual violence, the concept of "divine violence" adds a robust discursive impetus to how people understand violence as a meaning-making phenomenon. Divine violence, Benjamin (in Zizek 2008) wants us to believe, lies in the paradox that exists between what God wants and what, in actuality, happens in God's name. Divine violence is an antithesis of divine predicates; it is a transgressive understanding of the boundaries and borderless conception of violence in its divine–human relations. Accordingly, Benjamin argues that "Mythical violence is bloody power over mere life for its own sake, divine violence is pure power over all life for the sake of the living. The first demands sacrifice; the second accepts it" (cited in Zizek 2008, p. 198). Here, the command to commit violence and the prevention of violence consecutively take place. The turn and gap between commandment and prevention are what humans might wrestle with in their solitary confinement or ignore altogether. Zizek (2008, p. 198) elaborately drives home the concept of divine violence.

> It is this domain of pure divine violence which is the domain of sovereignty, the domain within which killing is neither an expression of personal pathology (idiosyncratic, destructive drive), nor a crime (or its punishment), nor a sacred sacrifice. It is neither aesthetic, nor ethical, nor religious (a sacrifice to dark gods). So, paradoxically, divine violence does partially overlap with the bio-political disposal of *Homini sacer*: in both cases, killing is neither a crime nor a sacrifice. Those annihilated by divine violence are fully and completely guilty: they are not sacrificed, since they are not worthy of being sacrificed to and accepted by God—they are annihilated without being made a sacrifice.

One might ask: what does this hold for violent prayers? Violent prayer, as we will shortly describe, puts the litigant in the gap between the paradox of "killing", in this instance, not physical killing, and yet not feeling guilt for the act of killing. The killed (victim) is fully and completely guilty of the "sin" of oppositionality, and his death is thus justified on the basis that the litigants experience both existential and perhaps material deliverance. When violent prayers, as we found out, are targeted at "the spirit of COVID-19" for instance, the death of that spirit (assuming the spirit dies) is justified because it is "a being that works against the purpose of God for humanity" and, therefore, "guilty of causing the COVID-19 pandemic" (Interview with Emmanuel 2021).[3] Zizek (2008, p. 199) again helps us to understand this interpretation more clearly when he posits in this context that "one can also kill without committing a crime and without sacrifice". Put literally, to kill without committing a crime as an act of divine violence means that the victim's death is justified on the basis of established guilt, and God cannot be questioned nor regarded as committing the crime of killing the victim; second, the death of the victim, as a consequence of sin committed and death that resulted from it, cannot be understood as a sacrifice to God. In Pentecostal circles where violent prayers are common, for instance, assuming such prayers are believed to have resulted in the death of an enemy, the prayer litigant cannot be accused by the state of committing murder, and the victim cannot be accepted by God as a saint.

In the Pentecostal context, divine violence can be appropriated and contextualised within its poetic and performance of violent prayers. The performance of divine violence is situated within a political or existential time and space of how humans interpret their actions as divine intervention to either call human attention to divine purpose or God executing divine judgment on humanity via human or natural instrument. This theological

gleaning resonates in most of the disasters that humans have experienced, for example, the Spanish flu, HIV and AIDS, 9/11, Ebola, and now COVID-19, where a divine violence theory has been put in place to explain that the moral despicability of the world has reached a "heavenly" crescendo, which deserves a form of divine intervention. Conspiracy theory explained as eschatology and the formula of divine response to human deeds tends to justify the outbreak of COVID-19, which Goshen-Gottstein (2020) refuted as rationally untenable. Historical analysis by Adam Mohr (2020) illustrates how a global pandemic instigates religious fervour among people of faith who choose to look outside the official channels of medical aid (or scientific authority) for survival.

In any case, divine violence becomes justifiable to either refute or justify the disaster. But in Pentecostal context of violent prayer wherein the Pentecostals want a divine violence to occur, it has a redemptive or liberational purpose, at least, for existential reasons; and in particular reference to Africa, this form of prayer is tailored against "spirits whose violent hungers are sated by humans" (Whitehead 2002, p. 1). These preying spirits are wicked, malignant and unappeasable in their determination to feed on humans. In Nigerian Pentecostalism, any spirit other than the Holy Spirit is regarded as an evil spirit in urgent need of exorcism. These spirits do not only inhabit inanimate objects, they are also capable of possessing humans and using them to perform nefarious activities against other humans. It is for this belief that deliverance ministries more often than not ensure that their converts are made to go through deliverance session. As Kalu (2008) explains, even people who have had contact with African Indigenous Churches before coming to some of the Pentecostal churches would be requested to undergo deliverance session let alone the adherents of African Indigenous Religion who become Christians converts.

The Pentecostal belief that all spirits are not of God strongly influences their ascription of spirit to things they believe are opposed to God's purpose for their human flourishing. It is from this perspective that one can broadly comprehend the theological conception of COVID-19 as a spirit whose mission is to satiate itself with human lives. Divine violence thus becomes known in the Pentecostal circle as supernatural intervention with the evidence of

> destroying powers and principalities the devil use (sic) to fight his [God's] children. God cause (sic) great violent (sic) to the enemies so that he can deliver his people. The case of Pharaoh who perished in the Red Sea is an example of divine violent (sic). Moses and the children of Israel cried to God for deliverance. It was a great cry because they were before (sic) the Red Sea and the army of Pharaoh. . .. Pharaoh died a violent death in the Red Sea. That is why the children of Israel were able to escape. That is also the battle cry against COVID. By the grace of God, we have overcome it (Interview with Olusola 2022).[4]

In violent prayers, there is the spirit of urgency for deliverance at play; it is a prayer that encapsulates one's whole being in its tri-form—body, spirit, and soul—which is expressed in poetic resonance. In my fieldwork, I have come to realise that the performative or poetic is a totalising part of the aggressive expression or what we might call the poetic current of violent prayer. What is meant here is that violent prayer calls for the immediacy of divine violence or divine intervention to cause, change, transform, rectify, or blow off an oppositional force in operation in a litigant's real or imagined situation. In this sense, one respondent defines violent prayer the following way:

> Violent prayers are prayers. . . number one, you don't pray them silently. You are not permitted to pray violent prayer silently. Violent prayers are prayers you pray with the spirit of desperation. Violent prayers are prayers you pray with aggression. Violent prayers are prayers you pray with the intention of getting what you want at all cost (Interview with Ayeni 2022).[5]

The divine being intervening in the natural and overruling the law of nature is the goal of the violent prayer litigants: they expect divine violence to occur as often as they pray their importunate prayers. The performative and the expectative conjoin to describe

the intensity or tonality of violent prayers. In addition, there is also the intentionality that directs the mindset of the litigant towards a target; this target is first and foremost imagined and brought into a "tangibly reachable" realm "in the eye of the mind where you can target it without fail". This aptly describes the belief that underwrites the "Die, die, die in the name of Jesus!", which forcefully resonates in the prayer format of the Mountain of Fires and Miracles Ministries. The command to the spirit to die in the name of Jesus is best understood by the violent prayer litigants as a poetic performative engagement: this imagined target of violent prayer is beyond the biological, sociological, material, and ecological, even though the militant language often resonates with their register.

This conceptualisation of violent prayer as a poetic performative engagement with the invisible-made-visible forces profoundly represents the theology of prayer that embodies the Mountain of Fire and Miracles Ministries' (MFMM's) imprecatory prayers during the COVID-19 pandemic. The next section examines this theology.

### 4. Violent Prayers at the MFMM during COVID-19

The Mountain of Fires and Miracles Ministries (MFMM) started as a prayer meeting in 1989 in the house of Dr. Daniel O. Olukoya, the founder. The meeting, summoned by Olukoya, a molecular geneticist from the University of Reading in the United Kingdom, later metamorphosed into the Mountain of Fires and Miracles Ministries in 1994 when the first Sunday service was held in Lagos, Nigeria. The MFMM self-identifies as a prayer ministry devoted to "aggressive" and "violent" prayer. On its website, the MFMM hones the epithet "do-it-yourself gospel ministry where your hands are trained to wage war and your fingers to fight", implying that members are trained in violent prayers and should be able to apply the principles and methods to themselves and others. This makes every member a militant, violent litigant who prays kinetically or performatively against spirits that are contesting their destinies.

In the numerous publications by Olukoya (up to 600 books and booklets, especially his *Prayer Rain* (Olukoya 2013)), the theme of spiritual warfare, which I prefer to refer to as violent prayers, runs through them. In an interview with Olukoya, he describes violent prayer as what the elite would refer to as a crude, uncivilised, primitive, war-mongering, destructive, ballistic form of prayer, targeted at the devil and his cohorts. Ironically, he believes that such a form of prayer may not make sense to those who claim to be enlightened until they are faced with spiritual challenges beyond their immediate knowledge, skill, or medical science. Olukoya claims that the church engaged in violent prayers against the COVID-19 pandemic, and the result can be verified in the number of cases of infection or death in the country in comparison with developed countries that relied almost exclusively on their scientific or medical knowledge. He argued that although the claim of divine intervention might not make sense to outsiders, that is, people who do not believe in the power of violent prayers, those who believe know the mysterious powers that worked in favour of the country and African continent at large. Put in Wariboko's (2020, p. ix) words, "It does not make sense, but it makes spirit," that is, if Olukoya's explanation does not make logical, conventional sense in a secular world, it does make sense to those who share the same spiritual faith as him.

In the absence and even now the availability of the vaccine, the COVID-19 pandemic, for the MFMM, is first and foremost a "spirit" and, therefore, requires a spiritual battle. According to ethnographers Oyelade and Akintunde (2022, pp. 106–7), "It is certain that the COVID-19 pandemic is a global challenge, Pentecostals in Southwest Nigeria, have never done without warfare prayer against the 'spirit' or demon of coronavirus, in spite of the lockdown, even though there have been responses to the global challenge of COVID-19 in various ways". This belief in the "spirit" of COVID-19 is borne out of African cosmology that holds that sickness, affliction, disease, and so forth are mostly caused by some supernatural forces. As Adiele (2007, p. 146) observes, "This involves the belief that demons are responsible for every manner of sickness that attacks a child of God. Therefore for total solution, the demon of malaria, demon of cancer, demon of small pox, and so

on could be 'cast out in the mighty Name of Jesus". Given Olukoya's background as a self-acclaimed victim of spiritual oppression as a young man, his church, the Mountain of Fire and Miracles Ministries, believes and teaches that almost everything has a spirit, and the malignant spirits must be dislodged by the name of Jesus. Matthews Ojo (2010), an African church historian, argues that the MFMM stupendously appropriates and magnifies African indigenous spirits, and Christians are advised to identify the spirits responsible for their problems. More recently, Richman (2020) also confirms the overarching belief that spiritual forces are responsible for human problems. Although Richman views this belief as exaggerated, he, however, argues that it forms a critical basis for the flourishing of African Pentecostalism, especially as it relates to deliverance churches, such as the Mountain of Fire and Miracles Ministries.

Although there are various controversies concerning the origin of the COVID-19 pandemic, as well as conspiracy theories (see Aluko 2022), its spiritual etiology resonates with the theology of prayer in the MFMM. According to Akadiri,[6] "whether or not medical science recognises the spiritual dimension of COVID-19 pandemic, the point that is clear is that sustained violent prayers have saved Africa from its consequences." Akadiri, a pastor in the MFMM, reveals that the church was engaged in combative, relentless prayer to deal with the root and spirit of the pandemic. "What prayer cannot do hardly exists, and COVID-19 had to yield in the face of intensive and ferocious and ballistic prayers rained with brimstones against the spirit of COVID-19".

Even those who want to explain away the divine violence dimension of COVID-19, according to Dare[7], would have to contend with the contradictions inherent in their predictions. Dare thus argues that it makes little or no sense to hold the scientific belief that Africa survived the pandemic despite the minimal preparations, abject poverty, and near-absence of medical facilities, which should have been the very reasons for the fulfilment of the predictions against Africa. He further makes the argument that the same poverty and then climate became the defaults that saved Africa from the pandemic. "Yet, if we tell them that prayers work, they won't believe because they do not believe in God that answers prayers." Molade presses it home more deeply when she volunteered the following:

"Look at when this Corona, COVID-19 came, it was tamed. I want to believe that there is a hand of God in that. Coronavirus could not spread the way it killed other people in different places. Some say it is our weather, who gave us that weather? Some say it is because of the prayer. Yes, who gave us that instrument that were (sic) able to curtail the evil of COVID-19? So, our prayers, our intercession is still the one keeping this country."[8]

The unexpectedness and uncertainty that the pandemic occasioned also stirred a more combative spirit of violent prayers. "You know, when something happens suddenly and it takes you unaware, you are shaken. Then as a prayer warrior, you step back and take position, and then launch an offensive into the spiritual realm where that force is coming from"[9] Positionality and oppositionality are crucial in deploying violent prayers as a spiritual weapon: "you don't have to be at the defensive. That is why the church took uncompromising, offensive stand in the battle against COVID-19. COVID-19 is a spiritual attack, and we have to deal with it spiritually"[10] (Interview with Molade 2022).

In an interview with Esther,[11] although the church fervently prayed against the COVID-19 pandemic, members were always "instructed and reminded to observe COVID-19 prevention protocols". Two years after the lockdown, Dr. Olukoya still instructed members of his church to wear their face masks to every Power Must Change Hand programme. In fact, the seats were still arranged with enough space apart to observe social distancing in September 2022 when I visited the church. Esther also believes that the pandemic was a spirit, but that humans were responsible for its outbreak and spread around the world. For her, even though Christians could be protected against the pandemic by reason of their prayers, it was unreasonable to ignore safety protocols.

But the question of the Pentecostal embodiment is raised, that is, how the members of the church organised themselves in corporate worship during the period of the lockdown. To be sure, Pentecostal embodiment expresses the social and (spi)ritual roles in corporate

social life (Wilkinson 2020; Richman 2020). The bodies are not just a biological or physical aspect or site of our being; they are intensely social and cultural and interact differently in different contexts. During the lockdown, megachurches such as the MFMM reactivated their online capacities and facilities through which they organised their worship. Family churches became vibrant and the embodied Pentecostal practice of worship in the house boomed by attracting not only the Pentecostals but also the mainline church members who blurred their denominational idiosyncrasies and warmly worshipped together, thus creating and exhibiting a strong spirit of bond and fellowship. For the MFMM, the ritual of violent prayer intensified at the family level, where emotions of worship burst in a rather concentrated manner. The emotional experience of worship at the family level served to animate an intense desire for bonding and healing a world plagued by the virus. This family unit was given a theological grounding by recourse to "where two or three people are gathered, God is present". This interpretation ignited a new spirit of hope and intensity of violent prayer, even though the yearning for the return of the large congregation did not altogether wane. As I observed, the conviviality embodied in the return of the church speaks of the power of group solidarity and the boost of faith for divine violence to violate the violation of normalcy caused by the pandemic. However, as Oyelade and Akintunde (2022) observed, such embodiment cannot be compared to the pre-COVID and post-COVID experiences.

## 5. Conclusions

I have argued that prayer becomes more urgent and intense during times of crisis. First, there is evidence that faith in divine intervention in human affairs rises when everything else seems not to work. During the pandemic, the astronomical rise in prayer globally justifies the belief that prayers can help to cope in times of crisis. Second, a particular existential confrontation results in the kind of prayer that is used to counter it. We have seen that there is a genre of prayer deployed by the MFMM—violent prayers—that believes that violent times, such as the COVID-19 pandemic, demanded divine violence to counter the violence of the pandemic. Third, the belief in divine violence as a form of power capable of destroying existential threats became intense. The theology of violent prayers at the MFMM, which the prayer litigants and our interlocutors grounded on divine intervention, has helped them to cope with the effects of the pandemic. Fourth, violent prayer in large or small units is embodied; the expressive mode of praying, the swinging of the bodies, and the stomping of the ground with audacious vocal rhythm make the body more than a site of understanding but also of constructing a Pentecostal theology of embodied violent prayer. The poetic performative participation gave them hope in the midst of the uncertainties during the pandemic. Since violent times require violent prayers, the MFMM, known for its violent eruption of the spiritual realm and causation, furiously engaged in violent prayers for their existential liberation from the imminent global catastrophe that COVID-19 threatened. However, there is a need to further interrogate how the divine violence concept of violent prayer as embodied by the MFMM becomes a perpetual practice rather than a once-for-all confrontation.

**Funding:** This research was funded by John Templeton Religion Trust/NAGEL Institute for the Study of World Christianity at Calvin University, Michigan, USA with Grant Number EAR600, and the APC was funded by the Research Institute of Theology and Religion, University of South Africa.

**Institutional Review Board Statement:** The study was conducted in accordance with the Declaration of Helsinki, and approved by the Institutional Review Board (or Ethics Committee) of ADEKUNLE AJASIN UNIVERSITY (230321ERC1 and 24 March 2021).

**Informed Consent Statement:** Informed and verbal consent was obtained from all subjects involved in the study.

**Data Availability Statement:** No new data were created or analyzed in this study. Data sharing is not applicable to this article.

**Conflicts of Interest:** The author declares no conflict of interest. The funding sponsors had no role in the design of the study; in the collection, analyses, or interpretation of data; in the writing of the manuscript, and in the decision to publish the results.

## Notes

1   Interview with Pastor Adedeji Akadiri was conducted on 18 June 2022 at Akure.
2   Interview with Sunday Ayeni, a member of MFMM in Lagos, 17 August 2021.
3   Interview with Bosede Emmanuel was conducted in Lagos on the 17 August 2021.
4   Interview with Pastor Stephen Olusola was conducted on 4 July 2022 at Akure.
5   See note 1 above.
6   See note 1 above.
7   Sunday Dare is a Senior Lecturer and member of MFMM. The interview was conducted on 23 June 2023 at Akure.
8   Interview with Emman Molade, a pastor with MMFM at Akure, 16 June 2022.
9   Interview with Solomon Segun, a member of MFMM at Akure, 16 June 2022.
10  See note 9 above.
11  Esther Tope is a youth leader in MFMM. This interview was conducted on 17 April 2022 at Akungba-Akoko, Nigeria.

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
