# Peer review of "“Violent Times Call for Violent Prayers”: “Divine Violence” during the COVID-19 Pandemic in the Mountain of Fire and Miracles Ministries, Nigeria"

_religions, doi:10.3390/rel15040471_

Round 1

Reviewer 1 Report

Comments and Suggestions for Authors

Dear Author/s, 

First of all, I would like congratulate you the article. 

I find it interesting. 

However, I suggest to make changes that would improve the structure and understanding of the paper:

1. There is no clear aim of the article. What is the scientific aim of this paper? What are the research hypotheses? What are the research questions?

2. I suggest to correct the language of the text to make it more scientific and less colloquial. I find some phrases not consistent with a scientific text, ex. "You researchers always don’t link violent prayer with colonial experience of violence.  You need to study history." 241/242 If it was a quote, it wasn't marked correctly.

3. The conclusion does answer any research question. It only summarizes the text. 

Author Response

Aim of the paper: The broad aim of this article is to examine how the members of the Mountain of Fire and Miracles Ministries (MFMM) understood and deployed the concept of divine violence in their violent prayer mode during the pandemic. Generally construed, divine violence, means that God is a sovereign being whose authority cannot be questioned. As such, when he (God) deplores violence to counter human or other spiritual violent intrusion into his creation, the victims of divine violence are neither regarded as saints nor is their death perceived as a sacrifice (Zizek 2008). In the context of this article, MFMM conceives of divine violence as supernatural intrusion and intervention in human existential crisis, which aims to destroy the spiritual forces perceived to be responsible for the outbreak and spread of the COVID-19 pandemic.

• On quotation marks: They have all been done.
• On conclusion: I have significantly reworked the conclusion.

I expressed my profound gratitude to the reviewers for their critical and helpful comments.

Reviewer 2 Report

Comments and Suggestions for Authors

The author observes that some Pentecostals use Matthew 11:12 to ground their practice of violent prayer theologically. They argue that since the kingdom is taken by violence, violent prayer is the instrument to achieve it. It is a form of combative and aggressively confrontational prayer targeted at spirits believed to negatively influence human existential progress or success. Dislodging or overcoming these spirits through violent prayers or spiritual warfare, the litigants think that they will be set free to enjoy material success. Violent prayer aims at disestablishing political and spiritual powers and authorities in order to establish the kingdom of God. In this sense, it is subversive at its core. While the Western theological context for prayer is directed towards the socio-causation of a specific situation, Africa differs because it ascribes the causation of crises to spiritual forces. It views prayers as a spiritual means of accessing power, turning the body into a channel to draw and exercise power. This power is the ability to transcend the limit of one’s human capacity.

When the COVID-19 pandemic confronted the health of people – in Africa, about 3.3 million people died and 1.2 billion became infected - Pentecostals found hope to live through the violent times by recourse to “violent prayers.” They prayed for the termination of the pandemic, a violent threat that demanded violent prayers. Pentecostal’s belief that all spirits are not of God strongly influences their ascription of things they believe are opposed to God’s purpose for their human flourishing to spirits. Hence, COVID-19 was viewed as the work of a spirit whose mission was to satiate itself with human lives. The Mountain of Fires and Miracles Ministries (MFMM) believed the result of the efficacy of their prayers could be verified in the number of 458 cases of infection or death in the country in comparison with developed countries that relied almost exclusively on their medical and scientific or medical knowledge.

The only remark is that the use of language requires some minor editing and referencing should be checked to be in order.

Comments on the Quality of English Language

None

Author Response

I have edited the entire article.

Reviewer 3 Report

Comments and Suggestions for Authors

The essay presents an important topic, which is the role of violent prayers during the COVID-19 pandemic in the Mountain of Fire and Miracles Ministries in Nigeria. It draws from relevant literature on Pentecostalism in Nigeria, literature on the study of violence, and qualitative research in the form of interviews. 

Although it is well-written and presented overall, the essay presents some problems with the structure and development of the argument. The central argument could be more explicit in the first section. Given that the title of the essay and one central aspect of it is the role of ‘divine violence’, it would be worth articulating clearly an argument about the role of divine violence in the kind of violent prayers described here. The poetic-performative quality of this kind of prayer is central to the discussion. It deserves more attention in how it is articulated and presented in the argument at the beginning of the article. 

There needs to be more discussion of other ethnographic studies on the role of prayer in Pentecostalism, especially in Africa. These are some examples of references that are not included:

Akintunde E. Akinade. (2020). “Holy Dilemma: Engaging Prayer and Power in African Pentecostalism.” Journal of World Christianity 12, 10 (2): 147–169. 

Bandak, A. (Ed.). (2021). The Social Life of Prayer: Anthropological Engagements with Christian Practice (1st ed.). Routledge. https://doi.org/10.4324/9781003149934, Chapters 16 and 22. 

Naomi Richman (2020). Machine gun prayer: the politics of embodied desire in Pentecostal worship, Journal of Contemporary Religion,35:3, 469-483, DOI: 10.1080/13537903.2020.1828506

Wilkinson, M. (2017). "Chapter 1 Pentecostalism, the Body, and Embodiment". In Annual Review of the Sociology of Religion. Leiden, The Netherlands: Brill. https://doi.org/10.1163/9789004344181_003

It would be important to situate this study and its argument in the context of the broader literature on prayer, embodiment and violence in Pentecostalism.   

The section on violence is rather long, and the discussion of authors such as Arendt could be more targeted to the specific study and argument. A condensed and more targeted discussion on violence could be presented earlier in the essay to frame the discussion and sharpen the argument on violent prayers and how these came to be understood in the context of the COVID-19 pandemic. 

In terms of the methodology, the essay mentions that the author used ethnographic methods such as interviews and participant observation. However, it would be worth mentioning the length of fieldwork and dates. The article provides enough evidence of interviews, but there needs to be an ethnographic description and evidence that the author did participant observation. It would be helpful to illustrate this descriptively. For example, how did the prayers change during the periods of lockdown during the COVID-19 pandemic? The author mentions some safety issues in place, such as social distancing at church, but what did this mean for the embodied power and violent prayers? What happened to this power in a context where bodies were not meant to mix? I would also assume that mass gatherings did not occur for some time. So, the reader needs to know what happened during the height of the pandemic. How did the violence of prayers manifest itself in practice? 

Regarding formatting, some interview extracts do not have quotation marks, which must appear as block quotations instead of paragraphs. For example,

 212-217, the quote from pastor Akadiri needs quotation marks or a blocked quotation format. 

Also, testimony from 241-250.

502-506 testimony needs block quotations.

411, there is a typo. 

Finally, line 518 should say face masks instead of nose masks.

Comments on the Quality of English Language

I've included all my comments above. 

Author Response

• Central argument: I have reworked the introduction and conceptualized divine violence in the introduction
• I have provided more information on the ethnographic studies
• I added Richman and Wilkson. I do not have active access to other resources suggested.
• I have shortened the section of violence
• I have provided information on what happened during the lockdown
• Quotation marks have been addressed and nose mask changed to face mask

I expressed my profound gratitude to the reviewers for their critical and helpful comments.

Round 2

Reviewer 1 Report

Comments and Suggestions for Authors

Thank you for the answer to my comments and congratulation for this text. 

Author Response

I expressed my profound gratitude for your critical and helpful comments.